# KCNE1 tunes the sensitivity of $K_V7.1$ to polyunsaturated fatty acids by moving turret residues close to the binding site

Johan E Larsson[1], H Peter Larsson[2], Sara I Liin[1]*

[1]Department of Clinical and Experimental Medicine, Linköping University, Linköping, Sweden; [2]Department of Physiology and Biophysics, University of Miami, Miami, United States

**Abstract** The voltage-gated potassium channel $K_V7.1$ and the auxiliary subunit KCNE1 together form the cardiac $I_{Ks}$ channel, which is a proposed target for future anti-arrhythmic drugs. We previously showed that polyunsaturated fatty acids (PUFAs) activate $K_V7.1$ via an electrostatic mechanism. The activating effect was abolished when $K_V7.1$ was co-expressed with KCNE1, as KCNE1 renders PUFAs ineffective by promoting PUFA protonation. PUFA protonation reduces the potential of PUFAs as anti-arrhythmic compounds. It is unknown how KCNE1 promotes PUFA protonation. Here, we found that neutralization of negatively charged residues in the S5-P-helix loop of $K_V7.1$ restored PUFA effects on $K_V7.1$ co-expressed with KCNE1 in *Xenopus* oocytes. We propose that KCNE1 moves the S5-P-helix loop of $K_V7.1$ towards the PUFA-binding site, which indirectly causes PUFA protonation, thereby reducing the effect of PUFAs on $K_V7.1$. This mechanistic understanding of how KCNE1 alters $K_V7.1$ pharmacology is essential for development of drugs targeting the $I_{Ks}$ channel.
DOI: https://doi.org/10.7554/eLife.37257.001

*For correspondence:
sara.liin@liu.se

## Introduction

The voltage-gated potassium channel $K_V7.1$ and the auxiliary subunit KCNE1 together form the slowly activating and voltage-gated $I_{Ks}$ potassium channel, an important channel for cardiomyocyte repolarization (*Nerbonne and Kass, 2005*). More than 300 mutations in the genes encoding for $K_V7.1$ and KCNE1 have been found in patients with cardiac arrhythmias (*Hedley et al., 2009*). Mutations that reduce $I_{Ks}$ currents delay repolarization of the ventricular cardiac action potential and prolong the QT interval in the electrocardiogram, referred to as Long QT syndrome (*Hedley et al., 2009*). Long QT syndrome is a known risk factor for ventricular fibrillation and sudden cardiac death (*Nerbonne and Kass, 2005*). Up to 30% of patients with inherited Long QT syndrome are not protected against severe cardiac events using current anti-arrhythmic treatments (*Goldenberg et al., 2006*; *Priori et al., 2004*). Therefore, several studies have promoted the need for novel pharmacological drugs that increase or even restore the function of mutated potassium channels critical for cardiomyocyte repolarization; as these drugs could potentially be used to treat Long QT syndrome in carriers with loss-of-function potassium channel mutations (*Anderson et al., 2014*; *Perry et al., 2016*).

Several promising compounds have been found to activate the $K_V7.1$ channel (*Busch et al., 1994*; *Gao et al., 2008*; *Mattmann et al., 2012*; *Salata et al., 1998*). Unfortunately, the effects of several $K_V7.1$ channel activators are dramatically impaired by KCNE1 (*Busch et al., 1997*; *Gao et al., 2008*; *Salata et al., 1998*; *Yu et al., 2013*). For example, we have previously described that the activating effect of polyunsaturated fatty acids (PUFAs) on the human $K_V7.1$ channel, expressed in *Xenopus* oocytes, is impaired by KCNE1 (*Liin et al., 2015b*). Because $K_V7.1$ is co-assembled with KCNE1 in

**eLife digest** The muscle cells in the heart must contract and relax in a coordinated way for the heart to pump blood efficiently around the body. Different ions flow in and out of these cells, which are known as cardiomyocytes, to control when they contract and relax. The ions enter and leave by passing through channel proteins in each cell's membrane. People with mutations in the genes that encode these channel proteins are often prone to irregular heartbeats and may suddenly die because their heart stops pumping blood effectively.

Drugs that can restore the function of mutated ion channels are considered an attractive new avenue for treating the irregular heartbeats and preventing the sudden deaths. Yet a poor understanding of how the other proteins that interact with these channels may reduce the effect of these drugs has hampered their development. In 2015, researchers showed that some polyunsaturated fatty acids could help restore a normal heartbeat via an effect on the IKs channel, one of the ion channels that regulate the electric activity in cardiomyocytes. But, a protein called KCNE1 – which forms part of the IKs channel – reduced the effect of these fatty molecules via an unknown mechanism.

Now, Larsson et al. – who are three of the researchers involved in the 2015 study – report how KCNE1 reduces the effect of polyunsaturated fatty acids on the IKs channel. The experiments involved mutated human IKs channels produced in the egg cells of African clawed frogs – a popular model system for a wide variety of biological studies. Larsson et al. found that flexible loop-like part of the IKs channel has an overall negative charge that attracts positively charged hydrogen ions to the polyunsaturated fatty acid. This masks the electrical change of the fatty acid so that it no longer has any effect on the IKs channel. Yet, this phenomenon only occurs when KCNE1 is present, suggesting that KCNE1 moves specific parts of the loop close to the polyunsaturated fatty acid.

Several ion channels in cardiomyocytes are made from multiple subunits. Understanding how some of these subunits alter the effect of drugs will help scientists to develop drugs that efficiently act on these ion channels. Such drugs may offer new treatments for irregular heartbeats and prevent the sudden deaths. But as with all new drugs, extensive testing and clinical trials will be needed before anything reaches the clinic.

DOI: https://doi.org/10.7554/eLife.37257.002

the native $I_{Ks}$ channel complex in the heart (*Barhanin et al., 1996*; *Sanguinetti et al., 1996*), $K_V7.1$ channel activators must affect the $K_V7.1$+KCNE1 complex (referred to as $K_V7.1$+E1) to prevent cardiac arrhythmias, such as in Long QT syndrome. Although KCNE1 is important for the pharmacology of the $I_{Ks}$ channel, little is known about the molecular mechanisms underlying how KCNE1 changes the sensitivity of $K_V7.1$ to various compounds. This lack of mechanistic understanding limits the clinical utility and further rational design of several $K_V7.1$ channel activators that potentially could be used to improve treatment of patients with conditions due to compromised $K_V7.1$+E1 channels.

$K_V7.1$, the alpha subunit of the $I_{Ks}$ channel, is a potassium channel protein composed of six membrane-spanning segments, S1-S6: Helices S1 to S4 form the peripheral voltage-sensing domains and helices S5 and S6 form the central pore domain (*Figure 1A–B*) (*Liin et al., 2015a*). KCNE1, a single-transmembrane protein, is proposed to interact with $K_V7.1$ in the lipid-filled space between two voltage-sensing domains (*Figure 1B*) (*Chung et al., 2009*; *Nakajo and Kubo, 2015*; *Xu et al., 2013*). We have previously proposed that PUFAs incorporate into the outer leaflet of the cell membrane in the same lipid-filled space as KCNE1, but they incorporate closer than KCNE1 does to the transmembrane segments S3 and S4 (*Figure 1B*) (*Liin et al., 2015b*). In this position close to S4, negatively charged PUFAs, such as docosahexaenoic acid (DHA), facilitate $K_V7.1$ channel opening by electrostatically promoting the outward movement of the positively charged S4 helix (*Figure 1C*) (*Liin et al., 2015b*). As the DHA is negatively charged, DHA shifts the voltage dependence of $K_V7.1$ channel opening toward more negative voltages (*Figure 1C*) (*Liin et al., 2015b*).

However, we previously observed that this activating effect of DHA at physiological pH (i.e. pH 7.4) was abolished when $K_V7.1$ was co-expressed with KCNE1 to form the $I_{Ks}$ channel complex (*Liin et al., 2015b*). In addition, we proposed that this reduced effect is the result of KCNE1 decreasing the local pH at the DHA-binding site, inducing protonation of the DHA carboxyl head at pH 7.4

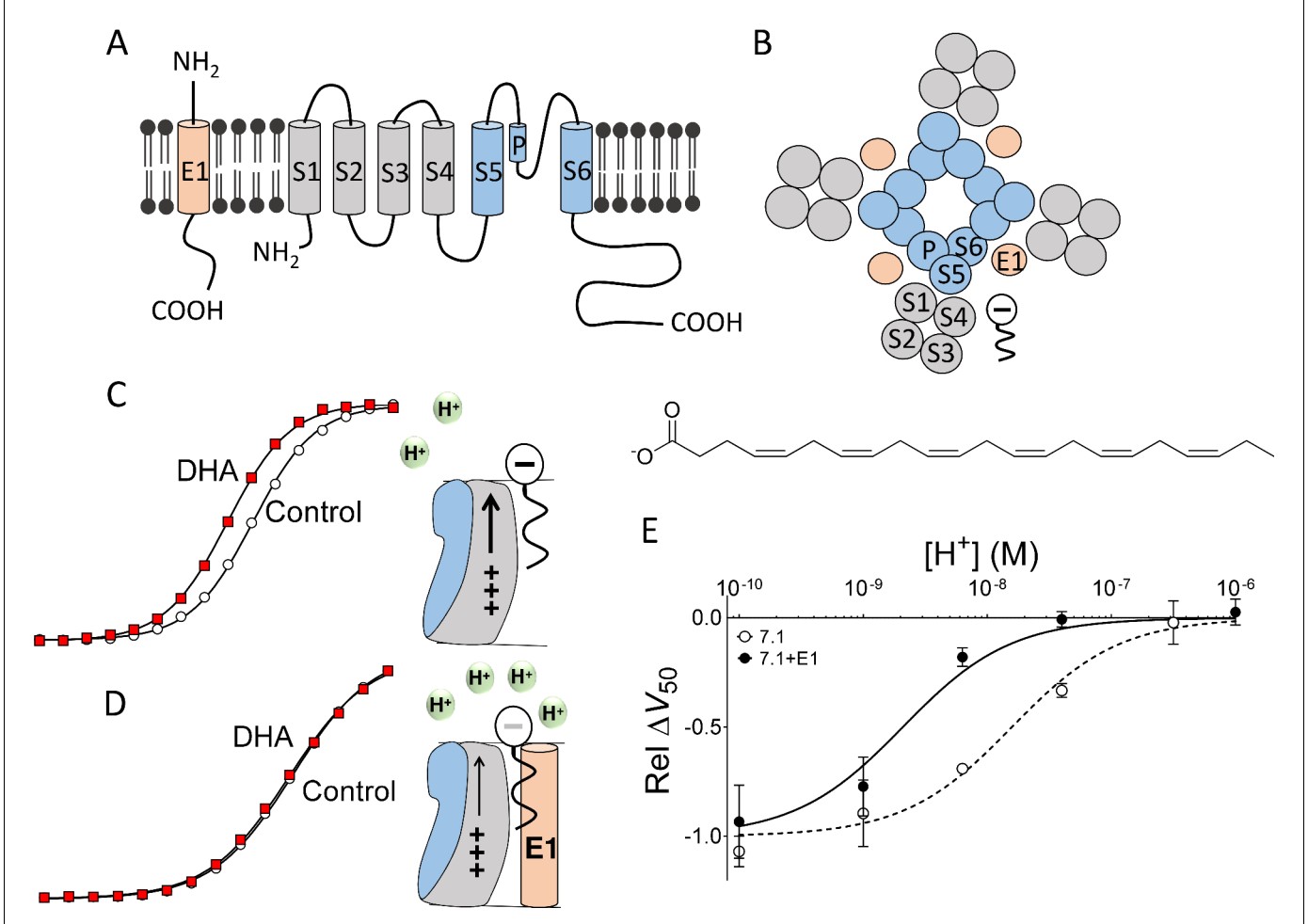

**Figure 1.** Concept of DHA-induced shift in $K_V7.1$ channel voltage dependence. (**A**) Schematic side view of one subunit of KCNE1 and $K_V7.1$. KCNE1 is in light orange. $K_V7.1$ is in grey (transmembrane helices S1-S4 forming the voltage-sensing domain) and blue (transmembrane helices S5 and S6 forming the pore domain). P denotes pore helix. (**B**) Schematic top-down view of the $K_V7.1+E1$ channel complex. Same coloring as in A. The putative localization of a polyunsaturated fatty acid between neighboring voltage-sensing domains is included. (**C**) Cartoon and representative example of previously published key data showing that at pH 7.4 70 µM DHA facilitates $K_V7.1$ channel opening by electrostatically facilitating outward S4 movement, seen as a shift of $G(V)$ curve towards more negative voltages (modified data from ***Liin et al. [2015b]***). (**D**) Cartoon and representative example of previously published key data showing that at pH 7.4, co-expression with KCNE1 decreases the local pH at the PUFA-binding site, which renders DHA uncharged and ineffective. As a consequence, 70 µM DHA fails to facilitate $K_V7.1+E1$ channel opening at pH 7.4, seen as a lack of shift of $G(V)$ curve toward more negative voltages (modified data from ***Liin et al. [2015b]***). (**E**) KCNE1 changes the pH dependence of the DHA-induced shift in the voltage dependence of channel opening, $\Delta V_{50}$. [DHA]=70 µM. The $\Delta V_{50}$ values were normalized to the fitted maximum $\Delta V_{50}$ (using ***Equation 2***) at very basic pH for each channel type to better visualize the different pKa values for the different channel constructs. Relative $\Delta V_{50}$ are expressed as mean ± SEM (modified and complemented data from ***Liin et al. [2015b]***). Best fit of ***Equation (2)***: pKa = pH 8.7 for $K_V7.1+E1$ and 7.8 for $K_V7.1$ alone. n = 3–6 per data point.

DOI: https://doi.org/10.7554/eLife.37257.003

(***Figure 1D***) (***Liin et al., 2015b***). Therefore, DHA becomes uncharged and ineffective at physiological pH. As a consequence, PUFA analogues with a lower pKa of the head group, which prevents protonation at physiological pH, was able to activate $K_V7.1+E1$ at physiological pH (***Liin et al., 2016***, ***2015b***). Moreover, we showed that the inhibiting effect of the positively charged PUFA analogue arachidonoyl amine (AA+) was potentiated by KCNE1, as if the decreased local pH at the PUFA-binding site further protonated the amine head of AA+ (***Liin et al., 2015b***). This improved protonation improves the electrostatic repulsion on the voltage sensor induced by AA+ (***Liin et al., 2015b***). However, it remains unclear how KCNE1 decreases the local pH at the PUFA-binding site. A mechanistic understanding of how KCNE1 tunes the pharmacology of the $I_{Ks}$ channel is critical for our

ability to predict which PUFAs modulate the $I_{Ks}$ channel, knowledge that will guide the development of synthetic PUFA analogues that pharmacologically target the $K_V7.1$+E1 channel. In this work, we propose a molecular mechanism that explains the KCNE1-induced protonation of PUFA.

To identify structural motifs in the $K_V7.1$+E1 channel that are responsible for PUFA protonation, we took advantage of the distinct pH dependence of the DHA effect on $K_V7.1$ and $K_V7.1$+E1. We previously described that the ability of DHA to shift the voltage dependence of $K_V7.1$ channel opening increases as pH increases, most likely due to deprotonation of DHA at higher pH (*Figure 1E*, dashed line) (*Liin et al., 2015b*). We also described that the pH dependence of the DHA effect on $K_V7.1$+E1 is shifted by about 1 pH unit compared to $K_V7.1$, making DHA completely protonated and ineffective on $K_V7.1$+E1 at physiological pH (*Figure 1E*, compare dashed and solid lines) (*Liin et al., 2015b*). In this work, we systematically mutated motifs in $K_V7.1$ and KCNE1 that could potentially cause KCNE1-induced protonation of DHA. We then compared the pH dependence of the DHA effect for each $K_V7.1$+E1 channel mutant to that of wild-type (WT) $K_V7.1$ with and without KCNE1 co-expressed. Our findings suggest that negatively charged amino acids in the S5-P-helix loop of $K_V7.1$ cause DHA protonation, but only when DHA is bound to the $K_V7.1$+E1 channel. We propose a model in which KCNE1 indirectly modulates the pharmacology of $K_V7.1$ by inducing structural rearrangements of the extracellular S5-P-helix loop of $K_V7.1$, moving acidic residues in this loop close to the DHA molecule.

## Results

### Removal of charged amino acids in KCNE1 did not affect the pH dependence of the DHA effect

First, we tested the impact that charged amino acids in the extracellular N terminus of KCNE1 have on the pH dependence of the DHA effect on $K_V7.1$. We created three KCNE1 constructs to systematically remove these charged amino acids. The first construct, E1/∆N2-38, removed most of the N

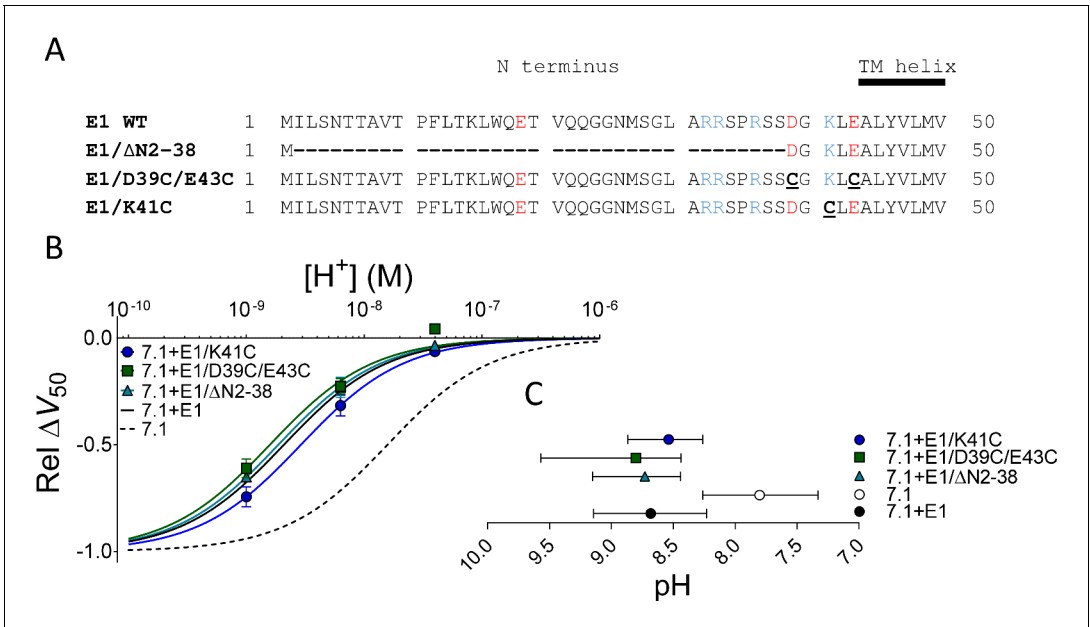

**Figure 2.** Removal of KCNE1 extracellular residues did not restore $K_V7.1$-like DHA effect. (**A**) Sequences of the N-terminal segments of WT human KCNE1 and three different KCNE1 mutants. Acidic residues colored red and basic residues colored blue. (**B**) pH dependence of the DHA effect (70 μM) on the relative $\Delta V_{50}$ for each KCNE1 mutant co-expressed with WT $K_V7.1$. Data shown as mean ± SEM. n = 3–6 per data point. Fits for $K_V7.1$ alone (black dashed line) and $K_V7.1$+E1 (solid black line) from data in *Figure 1E* are shown for comparison. (**C**) Apparent pKa for the DHA effect on indicated mutants. Data shown as mean ± asymmetrical 95% confidence interval determined using *Equation 2* on the data in B (best fit with *Equation 2*: pKa = pH 8.7 for $K_V7.1$+E1/∆N2-38, 8.8 for $K_V7.1$+E1/D39C/E43C, and 8.5 for $K_V7.1$+E1/K41C).
DOI: https://doi.org/10.7554/eLife.37257.004

terminus, including the charges E19, R32, R33, and R36 (*Figure 2A*). The second construct, E1/D39C/E43C, removed the two remaining negative charges in the N-terminal end of KCNE1, and the third construct, E1/K41C, removed the remaining positive charge (*Figure 2A*).

We assessed the pH dependence of the effect of extracellular application of 70 μM DHA (relative $\Delta V_{50}$, see Materials and methods for details) on these KCNE1 mutants to test whether each mutant had a $K_V7.1$+E1 like (continuous line in *Figure 2B*) or $K_V7.1$-like (dashed line in *Figure 2B*) pH dependence of the DHA effect. When co-expressed with WT $K_V7.1$, all three KCNE1 mutants generated currents with voltage dependence of channel opening shifted to more positive voltages compared to WT $K_V7.1$+E1 (*Supplementary file 1*). As shown earlier, interactions of the N-terminal end of KCNE1 with several parts of $K_V7.1$ (e.g. S1, S4, S6, and the S5-P-helix loop) may underlie the shifts in voltage dependence induced by these mutations (*Barro-Soria et al., 2017*; *Chung et al., 2009*; *Xu et al., 2013*). We found that the pH dependence of the DHA effect on all three constructs was similar to the pH dependence of the DHA effect on WT $K_V7.1$+E1 (*Figure 2B*). The apparent pKa of the DHA effect on WT $K_V7.1$ co-expressed with the KCNE1 mutants were close to the apparent pKa for the DHA effect on WT $K_V7.1$+E1 (*Figure 2C*). Thus, mutations of the extracellular N terminus of KCNE1 did not restore $K_V7.1$-like pH dependence of the DHA effect, as if extracellular charged amino acids in KCNE1 are not important for protonation of DHA in $K_V7.1$+E1.

## Removal of negative charges in the S5-P-helix loop affected the pH dependence of the DHA effect

Because charged amino acids in the N terminus of KCNE1 are not responsible for the KCNE1-induced change in the pH dependence of the DHA effect, we looked at charged amino acids in the extracellular loops of $K_V7.1$. The S5-P-helix loop in $K_V7.1$ is long and contains several negatively charged residues (*Figure 3A*). Xu *et al.* previously reported that cysteines introduced into the S5-P-helix loop of $K_V7.1$ form disulfide bonds with residues in the N terminus of KCNE1 (*Xu et al., 2013*). In other $K_V$ channels, the S5-P-helix loop exerts electrostatic effects on S4 due to its close proximity (*Broomand et al., 2007*; *Elinder et al., 2016*). Because the S5-P-helix loop could be in close proximity to the PUFA-binding site (which is proposed to be next to S4), we tested whether charged residues in the S5-P-helix loop influence DHA protonation. To this end, we created mutants in which the negatively charged amino acids E284, D286, E290, E295, and D301 in the S5-P-helix loop were, one by one, exchanged for cysteines (*Figure 3A*).

When co-expressed with WT KCNE1, four of the $K_V7.1$ mutants (D301C being the exception) generated currents with voltage dependence of channel opening that were shifted slightly to more positive voltages compared to WT $K_V7.1$+E1 (*Supplementary file 1*). By plotting the pH dependence of the DHA effect, we found that these mutants showed a range of pH-response curves in-between the curves of WT $K_V7.1$+E1 and $K_V7.1$ alone (*Figure 3B*). The pH-response curve for the DHA effect on $K_V7.1$/D301C + E1 most closely resembled the pH-response curve for the DHA effect on WT $K_V7.1$+E1 (*Figure 3B*, blue curve). In contrast, the pH-response curve for the DHA effect on $K_V7.1$/E290C + E1 overlapped with the pH-response curve for the DHA effect on $K_V7.1$ alone (*Figure 3B*, red curve). The apparent pKa of the DHA effect on $K_V7.1$/E290C + E1 was close to the apparent pKa for the DHA effect on WT $K_V7.1$ (*Figure 3C*). The apparent pKa of the DHA effect on $K_V7.1$/E290A + E1 or $K_V7.1$/E290C + E1 with DTT (1,4-Dithiothreitol) in the extracellular solution, to prevent formation of any potential disulfide bonds by E290C, were also similar to the apparent pKa for the DHA effect on WT $K_V7.1$ (*Figure 3—figure supplement 1A*). In addition, the apparent pKa of the DHA effect on $K_V7.1$/E290R + E1 was lower than for $K_V7.1$/E290A + E1 (*Figure 3—figure supplement 1B*). Altogether, these data suggest that negatively charged residues in the S5-P-helix loop (especially E290) promote the protonation of DHA in $K_V7.1$+E1. This protonation could be due to the negative charges in the S5-P-helix loop attracting hydrogen ions to the DHA-binding site. Because DHA protonation is promoted by KCNE1, a requisite for this hypothesis is that these negatively charged residues in the S5-P-helix loop are located close to the binding site for DHA when KCNE1 is present, but not when KCNE1 is absent. To further explore this possibility, we performed experiments using the $K_V7.1$/E290C mutation with the largest impact on the pH dependence of the DHA effect.

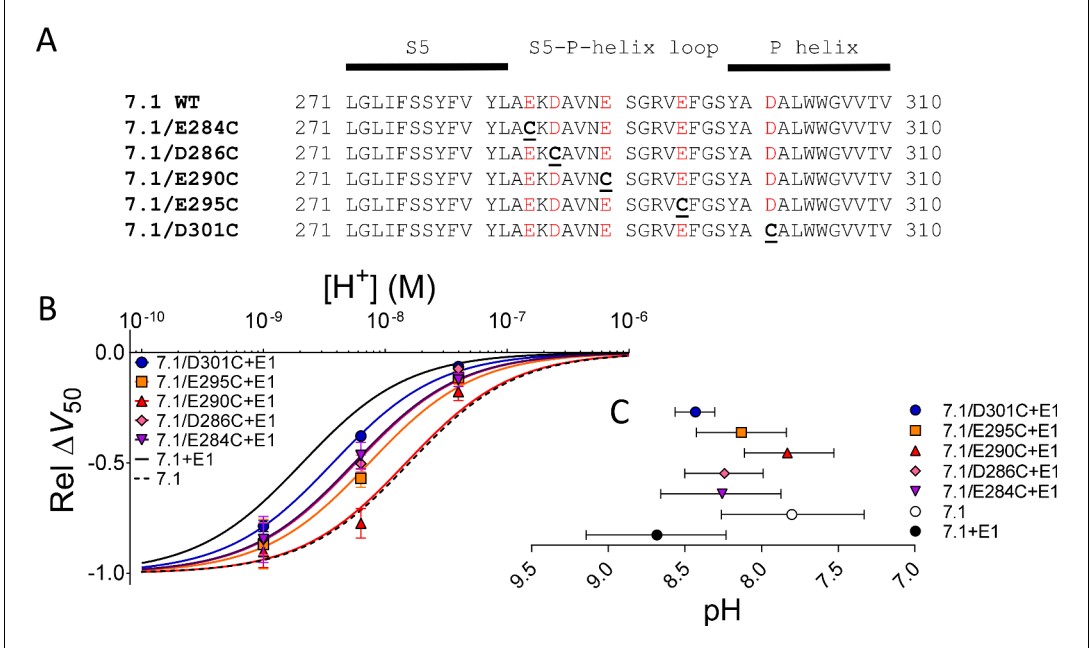

**Figure 3.** Removal of negative charges in the S5-P-helix loop partially or completely restored Kv7.1-like DHA effect. (**A**) Sequences of the S5-P-helix loop of WT human Kv7.1 and five Kv7.1 mutants. Acidic residues colored red. (**B**) pH dependence of the DHA effect (70 µM) on the relative $\Delta V_{50}$ for each Kv7.1 mutants co-expressed with WT KCNE1. Data shown as mean ± SEM. n = 3–7 for each data point. Fits for Kv7.1 alone (black dashed line) and Kv7.1+E1 (solid black line) from data in **Figure 1E** are shown for comparison. (**C**) Apparent pKa for the DHA effect on indicated mutants. Data shown as mean ± asymmetrical 95% confidence interval determined using **Equation 2** (best fit with **Equation 2**: pKa = pH 8.3 for Kv7.1/E284C + E1, 8.2 for Kv7.1/D286C + E1, 7.8 for Kv7.1/E290C + E1, 8.1 for Kv7.1/E295C + E1, and 8.4 for Kv7.1/D301C + E1). **Figure 3—figure supplement 1** is associated with **Figure 3**.

DOI: https://doi.org/10.7554/eLife.37257.005

The following figure supplement is available for figure 3:

**Figure supplement 1.** Alternative substitutions at position 290 support the hypothesis that the negative charge at position 290 is important for DHA protonation in the presence of KCNE1.

DOI: https://doi.org/10.7554/eLife.37257.006

## Neutralization of E290 did not affect the pH dependence of the DHA effect on Kv7.1 alone

To test the prediction that the E290C mutation does not alter the pH dependence of the DHA effect in the absence of KCNE1, we tested the effect of DHA on Kv7.1/E290C without KCNE1 co-expressed. As described previously (**Wang et al., 2015**), Kv7.1/E290C generated currents with WT Kv7.1-like voltage dependence for channel opening (**Supplementary file 1**). The pH-response curve for the DHA effect on Kv7.1/E290C alone closely resembled the pH-response curve for the DHA effect on WT Kv7.1 (**Figure 4A**, compare red dashed and black dashed curves). The apparent pKa of the DHA effect on Kv7.1/E290C was close to the apparent pKa for the DHA effect on WT Kv7.1 (**Figure 4B**). Extracellular application of 70 µM DHA at pH 8.2 shifted the voltage dependence of channel opening of Kv7.1/E290C similar to WT Kv7.1 (**Figure 4C**). This was clearly distinct from the altered pH dependence and increase in the DHA effect by the E290C mutation at pH 8.2 in the presence of KCNE1 (**Figure 4A–C**). These findings suggest that E290 only promotes DHA protonation when Kv7.1 is co-expressed with KCNE1.

## Restoring the negative charge at position 290 restored Kv7.1+E1 like pH dependence of the DHA effect

Next, we tested whether we could restore WT Kv7.1+E1 like response to DHA by restoring the negative charge at position 290 in the Kv7.1/E290C + E1 mutant. For these experiments, we used the negatively charged cysteine-specific sodium [2-sulfonatoethyl] methanethiosulfonate (MTSES⁻)

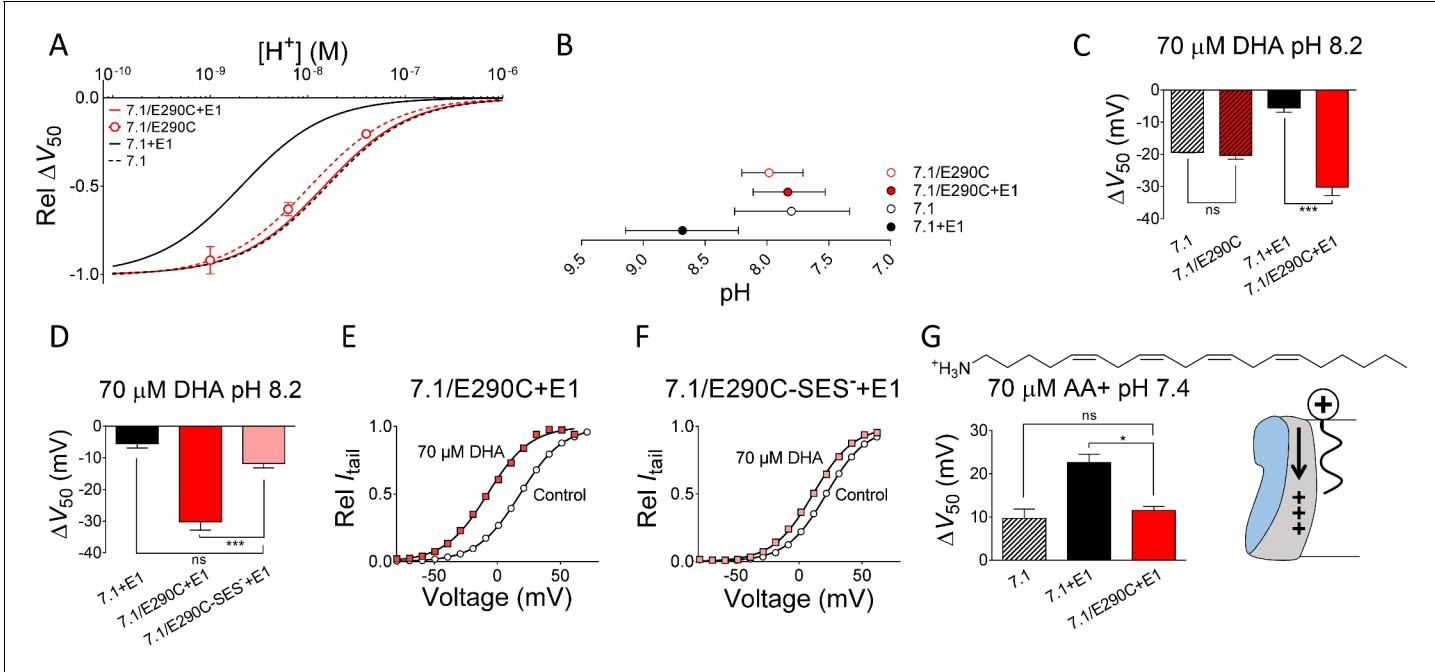

**Figure 4.** The charge at E290 is important for the PUFA effect. (**A**) pH dependence of the DHA effect (70 µM) on the relative $\Delta V_{50}$ for $K_V7.1$/E290C expressed without KCNE1. Data shown as mean ± SEM. n = 4–14 for each data point. Fits for $K_V7.1$ alone (black dashed line), $K_V7.1$+E1 (solid black line), and $K_V7.1$/E290C co-expressed with KCNE1 (red solid line) are shown for comparison. (**B**) Apparent pKa for the DHA effect on indicated mutants. Data shown as mean ± asymmetrical 95% confidence interval determined using **Equation 2** (best fit with **Equation 2**: pKa = pH 8.0 for $K_V7.1$/E290C). (**C**) Summary of $\Delta V_{50}$ induced by 70 µM DHA at pH 8.2 for indicated mutants. The E290C mutation only increases the DHA effect when KCNE1 is co-expressed. Data shown as mean ± SEM. n = 3–5. One-way ANOVA with Tukey's multiple comparison test. *** denotes p<0.001 and ns denotes p>0.05. (**D**) Summary of $\Delta V_{50}$ induced by 70 µM DHA at pH 8.2 for indicated mutants. MTSES⁻ modification of E290C is denoted by -SES⁻. MTSES⁻ modification of E290C restores WT $K_V7.1$+E1 like response to DHA. Data shown as mean ± SEM. n = 3–7. One-way ANOVA with Dunnett's multiple comparison test and $K_V7.1$/E290C-SES⁻+E1 set as control. *** denotes p<0.001 and ns denotes p>0.05. (**E–F**) Representative effect of DHA (70 µM) on the $G(V)$ curve at pH 8.2 for (**E**) $K_V7.1$/E290C + E1 and (**F**) $K_V7.1$/E290C-SES⁻+E1. Control data in black and DHA data in red. (**G**) Summary of effect of arachidonoyl amine (AA+, 70 µM, structure on top) on $V_{50}$ of indicated mutants at pH 7.4. The E290C mutation decreases the AA+ effect on $K_V7.1$+E1. Data shown as mean ± SEM (modified WT data from **Liin et al. [2015b]**). n = 3–10. One-way ANOVA with Dunnett's multiple comparison test and $K_V7.1$/E290C + E1 set as control. * denotes p<0.05 and ns denotes p>0.05. Schematic illustration describes electrostatic AA+-induced prevention of outward S4 movement (right). **Figure 4—figure supplement 1** is associated with **Figure 4**.

DOI: https://doi.org/10.7554/eLife.37257.007

The following figure supplement is available for figure 4:

**Figure supplement 1.** Lack of effect of MTSES⁻ modification on intrinsic properties of $K_V7.1$/E290C + E1.

DOI: https://doi.org/10.7554/eLife.37257.008

reagent to covalently attach the negatively-charged SES⁻ group to E290C. To maximize the chance of seeing a difference in the DHA effect, we compared the effect of DHA on $K_V7.1$/E290C + E1 with and without MTSES⁻ modification at pH 8.2, the pH at which the difference in the DHA effects was greatest between $K_V7.1$ and $K_V7.1$+E1. Modification of $K_V7.1$/E290C + E1 by extracellular application of 10 mM MTSES⁻ had no clear effect on the intrinsic properties of $K_V7.1$/E290C + E1 (**Figure 4—figure supplement 1**). However, modification of $K_V7.1$/E290C + E1 with MTSES⁻ dramatically reduced the ability of 70 µM DHA to shift $V_{50}$ (**Figure 4D–F**). The DHA-induced shift of $V_{50}$ in MTSES⁻ modified and unmodified $K_V7.1$/E290C + E1 was −11.8 ± 1.4 mV and −30.2 ± 2.6 mV, respectively. In addition, DHA induced a similar $V_{50}$ shift in both WT $K_V7.1$+E1 and MTSES⁻ modified $K_V7.1$/E290C + E1 at pH 8.2 (**Figure 4D**). This data further supports the notion that the negative charge at position 290 is important for tuning DHA protonation.

## Decreased effect of positively charged PUFA analogue arachidonoyl amine on $K_V7.1$/E290C + E1 compared to WT $K_V7.1$+E1

As a final test of whether E290 changes the local pH at the binding site of PUFAs, we tested the effect of arachidonoyl amine (AA+) on $K_V7.1$/E290C + E1. AA+ is a PUFA analogue in which the negatively charged carboxyl head has been exchanged for a positively charged amine head (structure in *Figure 4G*). We previously showed that AA+ shifts the $V_{50}$ of $K_V7.1$ and $K_V7.1$+E1 by approximately +10 and +23 mV, respectively (*Liin et al., 2015b*), as if KCNE1-induced protonation of the amine head improves the electrostatic repulsion on the voltage sensor induced by AA+ and further prevents channel opening (*Figure 4G*, cartoon). In the presence of KCNE1, here we found that mutation of E290 caused a significant reduction in the AA+-induced $V_{50}$ shift (*Figure 4G*, compare black and red bar). The AA+ effect on $K_V7.1$/E290C + E1 was similar to that on WT $K_V7.1$ alone (*Figure 4G*, compare red and striped bar), as if primarily E290 is responsible for the improved effect of AA+ in the presence of KCNE1. These experiments using the positively charged PUFA analogue AA+ provide further support for the hypothesis that E290 is important for the KCNE1-induced protonation of the PUFAs.

## Discussion

In this study, we examined how KCNE1 changes the pharmacology of $K_V7.1$ by inducing PUFA protonation. Our results show that negatively charged residues in the loop connecting S5 to the pore helix, but not charged residues in the extracellular part of KCNE1, are important for KCNE1-induced DHA protonation. Neutralization of residue E290 at the top of the turret in the S5-P-helix loop had the largest impact on DHA protonation. Neutralization of E290 fully restored $K_V7.1$-like pharmacological sensitivity of $K_V7.1$+KCNE1 to DHA. That is, DHA induced a shift in $V_{50}$ of $K_V7.1$/E290C + KCNE1 at pH 7.4 and the pH dependence of the DHA effect was similar as for $K_V7.1$ expressed without KCNE1. We further show that neutralization of E290 only improved the DHA effect on $K_V7.1$ when $K_V7.1$ was co-expressed with KCNE1 and that it was the negative charge at position E290 that was important for the change in PUFA effect. *Figure 5* shows our proposed model of how KCNE1 changes the pharmacology of $K_V7.1$ to PUFAs. We propose that KCNE1 indirectly promotes PUFA protonation by inducing conformational re-arrangements in the $K_V7.1$ channel, which moves the S5-P-helix loop closer to the PUFA-binding site. This hypothesis fits with the location of each tested amino acid and the size of the effect of each tested amino acid when neutralized: E290, the residue with the largest effect, is located in the middle of the long S5-P-helix loop and may easily reach over to the putative DHA binding site, whereas D301, the residue with the least effect, is in the P-helix, located far from the putative DHA-binding site (*Figure 5D*).

Although the structural details and extent of the KCNE1-induced re-arrangements in $K_V7.1$ will need more study, our proposed model agrees with previous findings. In a recently published cryo electron-microscopy structure of *Xenopus* $K_V7.1$, the S5-P-helix loop forms a negatively charged cap above the pore domain (*Sun and MacKinnon, 2017*). Especially S280 (which corresponds to E290 in human $K_V7.1$) reaches all the way to the ion-conducting pore (*Sun and MacKinnon, 2017*). This finding agrees with our proposed model in which the S5-P-helix loop is fairly far from the PUFA binding site in $K_V7.1$ expressed without KCNE1 (schematically illustrated in *Figure 5A*). When $K_V7.1$ was co-expressed with KCNE1, Xu *et al.* reported that cysteines introduced in the S5-P-helix loop of $K_V7.1$ (at positions 284, 286, 290, or 295) form disulfide bonds with cysteines introduced at positions 32 and 33 in the N terminus of KCNE1 (*Xu et al., 2013*). Chung *et al.* reported that cysteines introduced in the S5-P-helix loop of $K_V7.1$ (at positions 284, 285, or 286) may form disulfide bonds also with cysteines introduced at positions 40–43 in the very end of the N terminus of KCNE1 connecting to the transmembrane segment of KCNE1, especially for $K_V7.1$/E284C – E1/E43C and $K_V7.1$/D286C – E1/G40C (*Chung et al., 2009*). In addition, Y46 in the outermost end of the transmembrane segment of KCNE1 was found in molecular dynamics simulations to dynamically interact with residues G297-D301 in the S5-P-helix loop of $K_V7.1$ (*Xu et al., 2013*). These observations suggest that the S5-P-helix loop can reach all the way to KCNE1 in the lipid-filled space between neighboring voltage-sensing domains, a finding that agrees with our proposed model (schematically illustrated in *Figure 5B*). The charge distribution in the S5-P-helix loop will then determine to what extent protonatable compounds, such as PUFAs, are negatively charged and thereby able to electrostatically interact with the voltage sensor S4 (schematically illustrated in *Figure 5B–C*). It is, however,

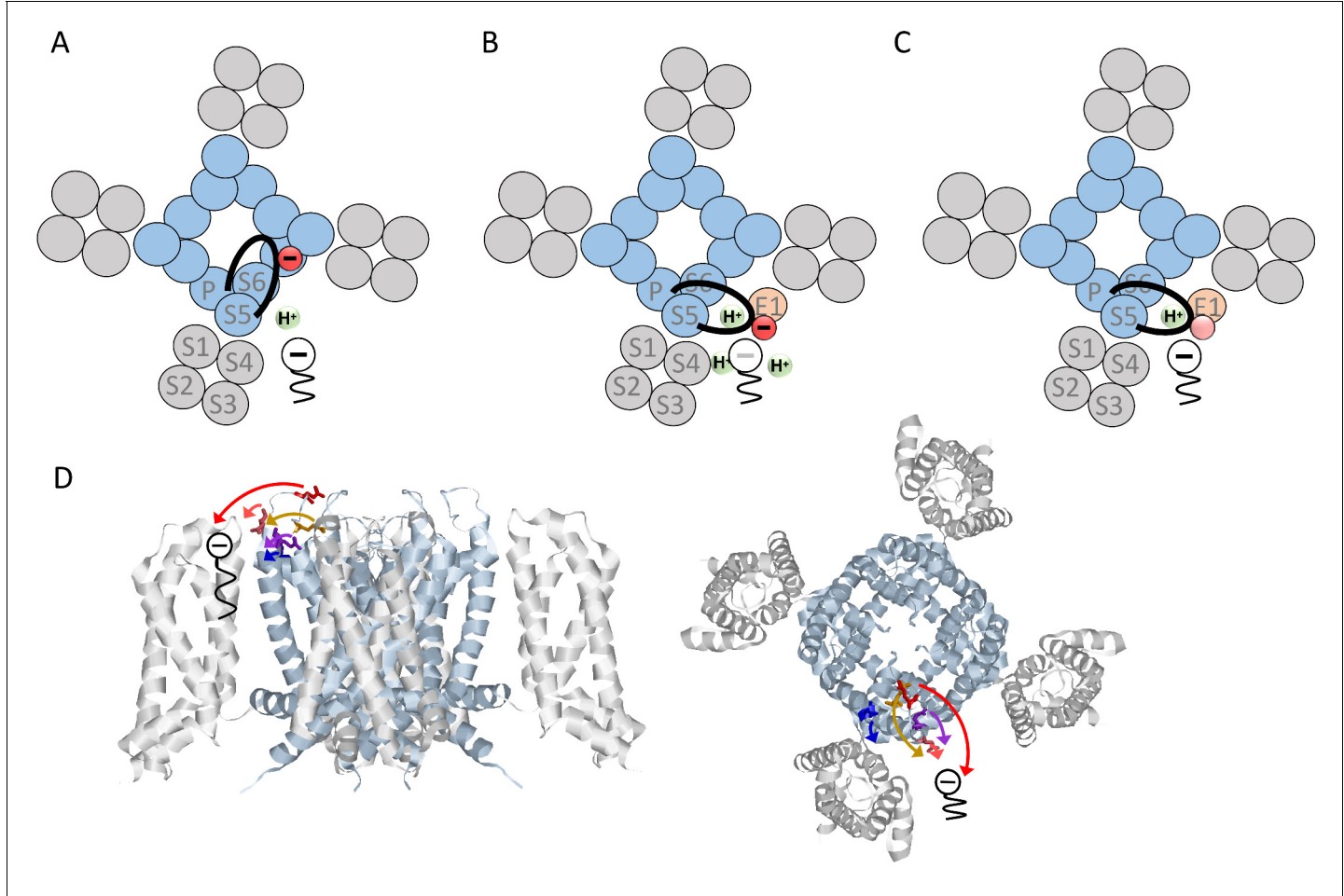

**Figure 5.** Proposed model of how KCNE1 induces protonation of PUFAs. (**A**) PUFA (black) binds to $K_V7.1$ in the lipid-filled space between two voltage-sensing domains (grey). The negatively charged PUFA head attracts the positively charges in S4 to facilitate channel activation. In $K_V7.1$ alone, the negatively charged S5-P-helix loop (black with red negative charge) belonging to the pore region (blue) is not close to the PUFA-binding site and will therefore not induce PUFA protonation. (**B**) KCNE1 (light orange) co-expression with $K_V7.1$ induces a conformational change that moves the S5-P-helix loop closer to the PUFA. The negative charges in the S5-P-helix loop, especially E290, attract protons to the PUFA binding site, which decreases the local pH. Decreased local pH causes PUFA protonation, which renders the PUFA uncharged and ineffective. (**C**) When negative charges in the S5-P-helix loop are neutralized, the ability of the S5-P-helix loop to attract protons is reduced. This tends to preserve the negative charge and the activating effect of the PUFA even in the presence of KCNE1. (**D**) Localization of indicated residues in the S5-P-helix loop of human $K_V7.1$, side view (left) and top-down view (right) (homology model based on the Cryo EM structure of *Xenopus* $K_V7.1$ [*Sun and MacKinnon, 2017*] and $K_V1.2/2.1$ [*Long et al., 2007*]). Arrows indicate the suggested translocation of the turret region towards the PUFA binding site. Same color coding as in *Figure 3* panel B and C (E284 in purple, D286 in pink, E290 in red, E295 in orange, and D301 in blue).

DOI: https://doi.org/10.7554/eLife.37257.009

insufficient to remove the N-terminal KCNE1 residues, as in our KCNE1/ΔN2-38 construct, or to neutralize charged residues in the N-terminus of KCNE1 to restore $K_V7.1$-like pH dependence of the DHA effect. We therefore find it unlikely that the S5-P-helix loop is attracted to KCNE1 by the N terminus of KCNE1. Instead, we propose that the binding of the KCNE1 transmembrane segment to the transmembrane segments of $K_V7.1$ induces a re-arrangement of $K_V7.1$, which moves the top of the turret (the S5-P-helix loop) and its acidic residues closer to the PUFA binding site; therefore these acidic residues in the S5-P-helix loop promote PUFA protonation.

During the last two decades, several $K_V7.1$ and $I_{Ks}$ channel activators have been identified (e.g. *Busch et al., 1994*; *Gao et al., 2008*; *Mattmann et al., 2012*; *Salata et al., 1998*). KCNE1 has a major impact, either positive or negative, on the effect of some of these activators. For example, ML277, ZnPy, and R-L3 activate $K_V7.1$, but the effect is reduced by KCNE1 co-expression (*Gao et al., 2008*; *Salata et al., 1998*; *Yu et al., 2013*). The proposed mechanism for the reduced

sensitivity in $K_V7.1+E1$ channels to these compounds is that KCNE1 and the compound compete for the same overall binding site (*Gao et al., 2008*; *Seebohm et al., 2003*) or that KCNE1 blocks the access to the compound binding site (*Xu et al., 2015*). In contrast, mefenamic acid and DIDS (4,4´-diisothiocyanatostilbene-2,2´-disulfonic acid) have effects on $K_V7.1$ expressed alone smaller than on $K_V7.1$ co-expressed with KCNE1 (*Busch et al., 1997*). For DIDS and mefenamic acid, amino acids at the top of the KCNE1 transmembrane segment (KCNE1 amino acid 39–43) are important for the effect, but there is no clear mechanism for how KCNE1 increases the effect of mefenamic acid and DIDS (*Abitbol et al., 1999*). Altogether, it is clear that KCNE1 can impair or promote the effect of $K_V7.1$ channel activators through diverse mechanisms. Our novel model explains how KCNE1 impairs the effect of negatively charged PUFAs on $K_V7.1$ by indirectly promoting PUFA protonation.

A detailed mechanistic understanding of how KCNE1 impairs the sensitivity of $K_V7.1$ to activators will enable rational drug design of compounds that circumvent KCNE1-induced impairment. For example, charged PUFA analogues and related compounds may be chemically optimized to preserve their charge or designed to bind to a slightly different site to evade protonation promoted by KCNE1. A mechanistic framework for the design of $I_{Ks}$ channel activators may open up new avenues for treating cardiac arrhythmias, such as Long QT syndrome.

## Materials and methods

### Key resources table

| Resource | Designation | Source | Identifiers | Additional information |
|---|---|---|---|---|
| Gene (*H. sapiens*) | KCNQ1 | NA | GenBank Acc.No. NM_000218 | |
| Gene (*H. sapiens*) | KCNE1 | NA | GenBank Acc.No. NM_000219 | |
| Chemical compound, drug | Docosahexaenoic acid (DHA) | Sigma | Cat#: D2534 | |
| Chemical compound, drug | Sodium [2-sulfonatoethyl] methanethiosulfonate (MTSES-) | Toronto Research Chemicals | Cat#: S672000 | |

### Molecular biology

$K_V7.1$ (GenBank Acc.No. NM_000218) in expression plasmid pXOOM and KCNE1 (NM_000219) in pGEM have been previously described (*Jespersen et al., 2002*; *Schmitt et al., 2007*). Mutations were introduced using site-directed mutagenesis (QuikChange II XL with 10 XL Gold cells, Agilent, CA). Newly mutated constructs were sequenced at the core facility at Linköping University to ensure correct sequence. cRNA was prepared using T7 mMessage mMachine transcription kit (Ambion/Invitrogen, CA). RNA concentration was quantified using spectrophotometry (NanoDrop 2000c, Thermo scientific, MA).

### *Xenopus laevis* oocyte experiments

*Xenopus* oocytes were surgically isolated at Linköping University or purchased from EcoCyte Bioscience (Castrop-Rauxel, Germany). Animal experiments were uppriven by the local ethics committee. Isolated *Xenopus* oocytes were injected with 50 nl RNA (each oocyte injected with 50 ng $K_V7.1$ RNA for expression of $K_V7.1$ alone or 25 ng $K_V7.1$ RNA and 8 ng KCNE1 RNA for co-expression of $K_V7.1$ +E1). The oocytes were incubated at 16°C for 2 to 3 days before performing two-electrode voltage clamp experiments. The two-electrode voltage clamp recordings were performed using a Dagan CA-1B Amplifier (Dagan, MN). Currents were filtered at 500 Hz and sampled at 5 kHz. The holding voltage was generally set to −80 mV. Activation curves were generally generated in steps between −80 and +80 mV in increments of 10 mV (3 s duration for $K_V7.1$ alone and 5 s duration for $K_V7.1$ +E1). The tail voltage was generally set to −20 mV. In experiments using arachidonoyl amine, a brief hyperpolarizing pulse (50 ms at −120 mV) was introduced between the activation step and tail step to relief channels from inactivation, as previously described (*Liin et al., 2015b*). The control solution contained 88 mM NaCl, 1 mM KCl, 15 mM HEPES, 0.4 mM $CaCl_2$, and 0.8 mM $MgCl_2$. pH was set to 7.4 using NaOH. When performing experiments at higher pH, pH was set the same day as the experiment using NaOH.

## Test compounds

4,7,10,13,16,19-*all-cis*-Docosahexaenoic acid was bought from Sigma-Aldrich (Stockholm, Sweden). Arachidonoyl amine was synthesized in house, as previously described (*Liin et al., 2015b*). Stock solutions of the compounds were prepared in 99.5% ethanol. Final test solution was prepared shortly before experiments. Previously, the effective concentration of PUFA has been shown to be 70% of the nominal concentration due to PUFA binding to the chamber walls (*Börjesson et al., 2008*). Here, the PUFA concentrations are the estimated effective concentration (i.e. 70% of the nominal concentration). Control solution was applied using a gravity driven perfusion system. Test compounds were added manually using a syringe, as previously described (*Börjesson et al., 2008*). The chamber was cleaned in-between each oocyte using albumin-supplemented control solution.

For MTS experiments, fresh MTSES$^-$ (sodium [2-sulfonatoethyl] methanethiosulfonate, Toronto Research Chemicals Inc., North York, Ontario, Canada) stock solution of 1 M was prepared on the day of recording. The stock solution was kept on ice. Final MTSES$^-$ solution (10 mM) was diluted immediately before application to each oocyte and applied using a pump (Harvard Apparatus MP II, CMA Microdialysis, Sweden) with a speed of 0.5 ml/min for 6 min.

For DTT experiments, 0.5 mM DTT (1,4-Dithiothreitol, Sigma-Aldrich, Stockholm, Sweden) was added to the incubation solution and the control solution to prevent disulphide-bond formation during incubation and experiment.

## Electrophysiological analysis

Electrophysiological analysis was performed in GraphPad Prism 6 and 7 (GraphPad Software Inc., CA). To quantify the voltage dependence for channel opening, tail currents were measured shortly after stepping to the tail voltage and plotted against the preceding activation voltage. A Boltzmann function was fitted to the data to generate the conductance *versus* voltage ($G(V)$) curve:

$$G(V) = A1 + (A2 - A1)/(1 + \exp(\frac{v_{50} - v}{s})),$$ (1)

where A1 is the minimal conductance, A2 the maximal conductance, $V_{50}$ the midpoint (i.e. the voltage at which the conductance is half the maximal conductance determined from the fit) and s the slope of the curve. The slope of the curve (s) was constrained to be equal for control and PUFA in each oocyte. The difference in $V_{50}$ induced by DHA in each oocyte (i.e. $\Delta V_{50}$) was calculated to quantify the shift in the voltage dependence for channel opening. In the figures, $G(V)$ curves have been normalized between 0 and 1 based on the fitted maximum conductance for clarity. For representative current traces, the current generated by a voltage step 20 mV more negative than $V_{50}$ was selected.

To plot the pH dependence of the DHA-induced shift in $V_{50}$ as a function of the H$^+$ concentration, the following concentration-response curve was fitted to the data:

$$\Delta V_{50} = \Delta V_{50,max}/(1 + (\frac{[H^+]_{50}}{[H^+]})^{-1}),$$ (2)

where $\Delta V_{50,max}$ is the maximal shift in $V_{50}$ and [H$^+$]$_{50}$ the H$^+$ concentration needed to cause 50% of the maximal shift in $V_{50}$. $\Delta V_{50}$ was then normalized between 0 and $-1$ for each mutant (referred to as relative $\Delta V_{50}$). The normalization is based on the fitted maximal value of $\Delta V_{50}$ from *Equation (2)*, set as $-1$. Massive cell leakage at pH 10 prevented us from quantifying the DHA effect at pH 10 for KCNE1 and K$_V$7.1 mutants. Therefore, the Hill coefficient of the concentration-response curves was constrained to $-1$ (as found for the DHA concentration-response curve for WT K$_V$7.1) to make the fits more robust. [H$^+$]$_{50}$ values were determined with asymmetrical 95% confidence interval in GraphPad Prism 7. [H$^+$]$_{50}$ and confidence interval were log-transformed to achieve apparent pKa values.

## Statistical analysis

Average values are expressed as mean ± SEM or mean ± 95% confidence interval (indicated in each figure legend). Statistical analyses were done using one-way ANOVA followed by a multiple comparison test. Dunnett's multiple comparisons test was used when comparing to defined reference data. Tukey's multiple comparisons test was used when testing all data against each other. p<0.05 was considered statistically significant.

## Acknowledgements

We thank Dr. Fredrik Elinder at Linköping University, and Dr. Rene Barro-Soria and Briana Watkins at University of Miami for their valuable comments on the manuscript. This work was supported by the National Institutes of Health (R01GM109762; R01HL131461), and the Swedish Society for Medical Research, the Swedish Research Council, Linköping University, the County Council of Östergötland, and the Lions Foundation.

## Additional information

### Competing interests

H Peter Larsson, Sara I Liin: Inventor of a patent application (#62/032,739) based on these results, which has been submitted by the University of Miami. The other author declares that no competing interests exist.

### Funding

| Funder | Grant reference number | Author |
| --- | --- | --- |
| National Institutes of Health | R01GM109762 | H Peter Larsson |
| National Institutes of Health | R01HL131461 | H Peter Larsson |
| Svenska Sällskapet för Medicinsk Forskning | | Sara I Liin |
| Vetenskapsrådet | | Sara I Liin |
| Linköping University | | Sara I Liin |
| County Council of Östergötland | | Sara I Liin |
| Lion Foundation | | Sara I Liin |

The funders had no role in study design, data collection and interpretation, or the decision to submit the work for publication.

### Author contributions

Johan E Larsson, Conceptualization, Data curation, Formal analysis, Writing—original draft, Writing—review and editing; H Peter Larsson, Conceptualization, Formal analysis, Funding acquisition, Writing—original draft, Writing—review and editing; Sara I Liin, Conceptualization, Data curation, Formal analysis, Funding acquisition, Writing—original draft, Writing—review and editing

### Author ORCIDs

Johan E Larsson http://orcid.org/0000-0003-3852-1015
H Peter Larsson http://orcid.org/0000-0002-1688-2525
Sara I Liin http://orcid.org/0000-0001-8493-0114

### Ethics

Animal experimentation: Animal experiments were performed in strict accordance with the recommendation of The Linköping Animal Ethics Committee at Linköping University (protocol #53-13 ).

### Decision letter and Author response

Decision letter https://doi.org/10.7554/eLife.37257.013
Author response https://doi.org/10.7554/eLife.37257.014

## Additional files

### Supplementary files
• Supplementary file 1.
DOI: https://doi.org/10.7554/eLife.37257.010
• Transparent reporting form
DOI: https://doi.org/10.7554/eLife.37257.011

### Data availability
All data generated or analysed during this study are included in the manuscript and supporting files (Supplementary File 1).

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
