## [Decision Letter]

Thank you for submitting your article "KCNE1 Tunes the Sensitivity of K_V_7.1 to Polyunsaturated Fatty Acids by Moving Turret Residues Close to the Binding Site" for consideration by *eLife*. Your article has been reviewed by three peer reviewers, including László Csanády as the Reviewing Editor and Reviewer #1, and the evaluation has been overseen by Richard Aldrich as the Senior Editor.

The reviewers have discussed the reviews with one another and the Reviewing Editor has drafted this decision to help you prepare a revised submission.

Summary:

This study builds on previous work by the authors that demonstrated activation (i.e., a leftward shift in the voltage-activation curve) of K_v_7.1 channels by polyunsaturated fatty acids (PUFAs). That effect, shown to reflect an electrostatic interaction between the negatively charged PUFA headgroups and the gating charge arginines which promote outward movement of the voltage sensor, was diminished by co-expression of KCNE1 (E1). Furthermore, E1 suppressed the PUFA effect on K_v_7.1 by promoting protonation of the PUFA headgroups, likely by lowering local pH. Correspondingly, N-arachidonoyl taurine, a PUFA analog with a permanent negative charge, was efficacious in activating the native K_v_7.1/KCNE1 complex, and restored normal action potential duration in animal models of arrhythmia. PUFAs are of great interest in pharmaceutical development because their chemistries are highly manipulatable and they bind to less conserved regions of ion channels, suggesting the potential for channel-specific modulation and fewer off-target effects. In the future variants of such compounds might be useful as therapeutic agents and understanding the basis for PUFA activation will be important.

The present study addresses the mechanism of the E1-induced reduction in local pH around the PUFA headgroups. Using site-directed mutagenesis the authors demonstrate that charged extracellular residues of E1 are not involved in the process. In contrast, neutralization of negatively charged side chains in the K_v_7.1 turret loop abolish the effect of co-expressed E1 on the apparent pKa of PUFAs, whereas PUFA-stimulation itself of K_v_7.1 remains unaltered. The authors suggest that the presence of E1 alters the conformation of the turret loop in a way that makes its charged residues move closer to the PUFA binding site, allowing them to reduce local pH by attracting protons. This is an interesting study which provides a hint for how the presence of the KCNE1 subunit can alter the pharmacology of the cardiac IKs current, a question with obvious physiological relevance.

Essential revisions:

All three of us share a single concern which we request the authors address. We are concerned about the assumption that substituting a cysteine for a charged residue is the same as removing the charged side chain. One concern is the likely (but potentially variable) oxidizing environment of the different extracellular surface positions targeted, which could promote intersubunit disulfide bond formation in the mutants, thereby changing turret conformation and/or mobility rather than simply reducing surface charge. (E.g., in the homology model shown in Figure 5 the E290 side chains seem quite close to each other. Are the effects of the mutations also seen in the presence of extracellular DTT?) A second concern is the fact that the assay involves changes in pH, which could change the ionization status of the cysteine thiols. Thus, the chemistry of the thiols might render the cysteine less inert than, say, an alanine. To confirm the suggested electrostatic nature of the E290 – PUFA interaction, we ask the authors to perform the following experiments, and to add the results to Figure 4A-G:

1) Test an alanine substitution (E290A) as an alternative means of charge-neutralization. If their hypothesis is correct, this mutant should phenocopy the E290C mutant with respect to DHA response and pH dependence.

2) Test a charge reversal substitution at position 290. This could be done either using an E290R mutant, or by modification of E290C with MTSET^+^.

All these experiments should be attainable in two months of time.

---

## [Author Response]

Essential revisions:All three of us share a single concern which we request the authors address. We are concerned about the assumption that substituting a cysteine for a charged residue is the same as removing the charged side chain. One concern is the likely (but potentially variable) oxidizing environment of the different extracellular surface positions targeted, which could promote intersubunit disulfide bond formation in the mutants, thereby changing turret conformation and/or mobility rather than simply reducing surface charge. (E.g., in the homology model shown in Figure 5 the E290 side chains seem quite close to each other. Are the effects of the mutations also seen in the presence of extracellular DTT?) A second concern is the fact that the assay involves changes in pH, which could change the ionization status of the cysteine thiols. Thus, the chemistry of the thiols might render the cysteine less inert than, say, an alanine. To confirm the suggested electrostatic nature of the E290 – PUFA interaction, we ask the authors to perform the following experiments, and to add the results to Figure 4A-G:1) Test an alanine substitution (E290A) as an alternative means of charge-neutralization. If their hypothesis is correct, this mutant should phenocopy the E290C mutant with respect to DHA response and pH dependence.2) Test a charge reversal substitution at position 290. This could be done either using an E290R mutant, or by modification of E290C with MTSET^+^.All these experiments should be attainable in two months of time.

We agree with the reviewers that the chemistry of the cysteine increases the risk that the experiments may be influenced by disulphide bond formation or ionization of the cysteine thiol. To eliminate that risk, we have now included experiments using the E290A mutant, as an alternative means of charge neutralization, and DTT together with the E290C mutant, to prevent disulphide-bond formation. The pH dependence of the DHA response for both E290A+E1 and E290C+E1 in DTT largely overlay with the pH dependence of the DHA response for Kv7.1 alone and E290C+E1 without DTT. These new experiments lend further support to the hypothesis that it is the lack of the negative charge on E290 that removes the effect of KCNE1 on DHA protonation.

We have also performed experiments using the E290R mutant, to test if introduction of a positive charge at position 290 can further promote deprotonation of the DHA head. We find that the apparent pKa of the DHA effect on K_V_7.1/E290R+E1 is lower than for K_V_7.1/E290A+E1, which suggests that a positive charge at position 290 further shifts the pH dependence of the DHA response towards lower pH values.

We feel that these three sets of new control data mainly contribute to the interpretation of the data presented in Figure 3. Therefore, these new data are presented as Figure 3—figure supplement 1A-B.